# Risk Factors Associated with Hepatitis C Subtypes and the Evolutionary History of Subtype 1a in Mexico

**DOI:** 10.3390/v16081259

**Published:** 2024-08-06

**Authors:** Saul Laguna-Meraz, Alexis Jose-Abrego, Sonia Roman, Leonardo Leal-Mercado, Arturo Panduro

**Affiliations:** 1Department of Genomic Medicine in Hepatology, Civil Hospital of Guadalajara, “Fray Antonio Alcalde”, Guadalajara 44280, Jalisco, Mexico; s.laguna.meraz@gmail.com (S.L.-M.); alexisjoseabiology@gmail.com (A.J.-A.); sonia.roman@academicos.udg.mx (S.R.); leonardo.leal3084@alumnos.udg.mx (L.L.-M.); 2Health Sciences Center, University of Guadalajara, Guadalajara 44340, Jalisco, Mexico; 3Doctoral Program in Molecular Biology in Medicine, Health Sciences Center, University of Guadalajara, Guadalajara 44340, Jalisco, Mexico

**Keywords:** hepatitis c virus, risk factors, evolution, IDU, HCV 1a

## Abstract

The Hepatitis C Virus (HCV), with its diverse genotypes and subtypes, has significantly impacted the health of millions of people worldwide. Analyzing the risk factors is essential to understanding the spread of the disease and developing appropriate prevention strategies. This study aimed to identify risk factors associated with HCV subtype transmission and calculate the emergence time of subtype 1a in Mexico. A cross-sectional study was conducted from January 2014 to December 2018, involving 260 HCV-infected adults. HCV infection was confirmed via Enzyme-Linked Immunosorbent Assay, and viral load was measured by real-time PCR. Genotyping/subtyping tools were the Line Probe Assay and Sanger sequencing of the non-structural region 5B (NS5B). The most frequent HCV subtype was 1a (58.5%), followed by subtypes 1b (19.2%), 3a (13.1%), 2b (5.4%), 2a/2c (2.7%), 2a (0.8%), and 4a (0.4%). Intravenous drug use and tattoos were significant risk factors for subtypes 1a and 3a, while hemodialysis and blood transfusion were linked with subtype 1b. For the evolutionary analysis, 73 high-quality DNA sequences of the HCV subtype 1a NS5B region were used, employing a Bayesian coalescent analysis approach. This analysis suggested that subtype 1a was introduced to Mexico in 1976, followed by a diversification event in the mid-1980s. An exponential increase in cases was observed from 1998 to 2006, stabilizing by 2014. In conclusion, this study found that HCV subtypes follow distinct transmission routes, emphasizing the need for targeted prevention strategies. Additionally, the findings provide valuable insights into the origin of HCV subtype 1a. By analyzing the history, risk factors, and dynamics of the HCV epidemic, we have identified these measures: limiting the harm of intravenous drug trafficking, enhancing medical training and infrastructure, and ensuring universal access to antiviral treatments. The successful implementation of these strategies could lead to an HCV-free future in Mexico.

## 1. Introduction

Hepatitis C Virus (HCV) remains a significant global health problem, affecting 50 million people worldwide and causing 242,000 deaths annually due to cirrhosis and hepatocellular carcinoma [1]. Currently, HCV is classified within the *Flaviviridae* family and the *Hepacivirus* genus. Based on genomic differences, HCV is divided into eight genotypes and approximately 93 subtypes, named by numbers and letters [2]. The most common subtype of HCV worldwide is genotype 1, particularly subtypes 1a and 1b [3]. One of the most critical regions of the HCV genome is the non-structural region 5B (NS5B), which encodes an RNA-dependent RNA polymerase [4]. Due to its variability among different strains, this region has been commonly used in phylogenetic studies and for classifying HCV genotypes/subtypes [2,5].

The history of HCV began in modern times when, in 1975, Dr. Alter and colleagues discovered a high prevalence of hepatitis cases that could not be attributed to hepatitis A or B and classified them as non-A or non-B hepatitis [6]. At that time, this term was used globally, including Mexico, to describe a significant portion of hepatitis cases with an unknown etiology [7]. Before the 1970s, the prevalence of post-transfusion hepatitis was estimated to be 33% [8]. In Mexico, this would be equivalent to approximately 15.75 cases per 1000 units transfused [9]. This problem became even more serious in 1974 with the commercial trade of blood products. By 1986, it was estimated that there were between 700,000 and 800,000 blood transfusions in Mexico, with approximately a third being remunerated [10]. In 1987, to reduce the risk of post-transfusion hepatitis and the human immunodeficiency virus (HIV) epidemic, the commercialization of blood donations was prohibited in Mexico [10]. In 1989, the HCV genome was characterized, enabling the upcoming era of molecular biology diagnostics, and a specific antibody test was developed [10] that made it possible to conclude that HCV was the primary cause of non-A and non-B hepatitis worldwide [11].

The efforts to avoid the spread of HCV infections among the Mexican population have been implemented gradually. For example, the mandatory detection of anti-HCV antibodies, hepatitis B (HBsAg), and other blood-borne diseases in national blood banks to secure safe blood donations was instituted by law (Norma Oficial Mexicana NOM 003/SSA2/1993) in 1993 [12]. Further progress was achieved through the first molecular diagnostic tests carried out in the mid-1990s by our research laboratory [13] and later in other localities [14], establishing the first mapping of the molecular epidemiology of HCV in Mexico. In these studies, HCV subtype 1a was the most common in West Mexico, while subtype 1b was predominant in the remaining states. HCV infections were highly prevalent in women with antecedents of obstetric surgery-related blood transfusions before 1994 [15]. Updated molecular epidemiology studies have revealed a reduction in the prevalence of HCV 1b and the emergence of subtype 3a [16,17]. However, nationwide studies are required to expand the phylogeography of the major HCV subtypes circulating among the different risk groups [18]. Additionally, in the late 2000s, nucleic acid testing (NAT) was introduced in several central blood banks nationwide, enhancing the security of blood transfusions [19,20].

Finally, current interferon-free antiviral regimens using direct-acting agents for people living with HCV have not optimally reached the best levels of coverage throughout the health system due to the lack of systematic screening and administrative obstacles [21]. Therefore, HCV infection persists across Mexico, notably in the US–Mexico border states like Baja California Norte and Tamaulipas, as well as in the western states such as Jalisco and Sinaloa and the central regions including Mexico City and the State of Mexico [18]. Over the last decades, these new infections may be due to the emergence of novel and distinct risk factors associated with HCV transmission, which include, in general, injection drug use (IDU), imprisonment, blood transfusion, tattoos, sex work, surgical or dialysis procedures, and early sexual activity [22]. However, whether these risk factors are consistent across the main HCV subtypes circulating in Mexico remains unclear. Thus, this study aimed to identify the risk factors associated with the transmission of HCV subtypes and estimate the time of emergence of subtype 1a in Mexico.

## 2. Materials and Methods

### 2.1. Study Population and Design

A cross-sectional study was conducted from January 2014 to December 2018 at the Department of Genomic Medicine in Hepatology, Civil Hospital of Guadalajara, Mexico. Eligible patients were adults (>18 years old) referred to the Department for HCV confirmatory diagnostic testing. This study was explained to each participant before signing a written informed consent. Risk factors and general demographic data were collected through a medical interview using a structured questionnaire. HCV infection was assessed using the ELISA (Enzyme-Linked Immunosorbent Assay) method (AxSYM^®^, Abbott Laboratories, Chicago, IL, USA), and HCV viral load was tested by real-time PCR (COBAS^®^ AmpliPrep/COBAS^®^ TaqMan^®^ HCV Test, v2.0, Basel, Switzerland).

This study used two methodological approaches, as shown in Figure 1. In conjunction, 79 samples were tested with LiPA and 115 with Sanger sequencing. An additional 66 samples from a prior HIV/HCV cohort were included [23], bringing the total to 260 HCV samples.

### 2.2. RNA Isolation and Reverse Transcription

After confirming HCV infection, total RNA was extracted using a column method (QIAamp Viral RNA Kit, Qiagen Science, Hilden, Germany). The extracted RNA was then reverse transcribed into complementary DNA (cDNA) in a two-step process involving random primers and reverse transcriptase, following the manufacturer’s instructions (SuperScript III Reverse Transcriptase Kit, Thermo Fisher Scientific, Carlsbad, CA, USA).

### 2.3. HCV Genotyping/Subtyping

The VERSANT^®^ HCV Genotype 2.0 Assay (LiPA) (Siemens Healthcare Diagnostics, Tarrytown, NY, USA) was initially used to determine HCV genotypes/subtypes. Samples that LiPA could not subtype were analyzed using Sanger sequencing as described previously [21], which involved PCR amplification of the NS5B region, visualization on an agarose gel, purification, and sequencing with BigDye Terminator v3.1 chemistry (Applied Biosystems, Foster City, CA, USA). Fragments were analyzed via capillary electrophoresis using the 3130 Genetic Analyzer. HCV genotypes were identified by phylogenetic analysis using ClustalW for sequence alignment and the Maximum Likelihood method for evolutionary analysis with a bootstrap of 100 repetitions using MEGA software v11.0.13 (Appendix A: Phylogenetic Tree).

### 2.4. Bayesian Coalescent Analysis of HCV Subtype 1a

For the evolutionary analysis, 73 high-quality DNA sequences of the HCV subtype 1a NS5B region (average length 348 bp) were selected based on collection date availability (Appendix A: Alignment HCV database). Multiple sequence alignment and nucleotide substitution model selection were performed using MEGA software v11.0.13. The Tamura-Nei (TN93) model with four gamma rate categories was selected. The analysis of the temporal signal was conducted using Tempest v1.5.3. Bayesian coalescent skyline analysis was conducted using strict and relaxed molecular clocks, implemented in BEAUti v1.10.4 and run in BEAST v1.10.4. Model selection was based on the Akaike Information Criterion (AIC). 

For molecular clock calibration, the substitution rate was estimated at 3.074 × 10^−3^ substitutions per site per year. Strict and relaxed molecular clocks were evaluated by comparing the Effective Sample Size (ESS) results, with the uncorrelated lognormal relaxed molecular clock (UCLD) performing best. Two independent Markov Chain Monte Carlo (MCMC) chains were run, each including 500 million iterations, ensuring that the ESS values exceeded 200 for each evolutionary parameter. The convergence of the chains was monitored using Tracer v1.6. The AIC with bootstrap resampling with 1000 replicates and the relaxed model showed the best parameters.

The phylogenetic tree was constructed using the Maximum Clade Credibility (MCC) method in TreeAnnotator v1.10.4, discarding the first 10% of iterations as burn-in (Appendix A: Phylogenetic Tree HCV 1a raw data). The time to the most recent common ancestor (TMRCA) was the mean, with a 95% highest posterior density (HPD) interval. All HCV subtypes detected by Sanger DNA sequencing have been deposited in GenBank with accession numbers PP926325–PP926438.

### 2.5. Statistical Analysis 

In the statistical analysis, continuous variables were represented by their median and interquartile range (IQR), while categorical variables were presented as frequencies or percentages. Comparisons of categorical variables were conducted using the χ^2^ test or Fisher’s exact test, depending on applicability. The Shapiro–Wilk test was employed to assess data normality. Non-parametric continuous variables were compared using the Mann–Whitney test. Risk factors were calculated through univariate logistic regression analysis expressed as Odds Ratio with a 95% confidence interval (95% CI). Statistical analyses were performed using the Statistical Program for Social Sciences software (SPSS 22.0, IBM, Inc., Armonk, NY, USA). A significance level of *p* < 0.05 was considered statistically significant for all tests.

### 2.6. Ethics

This research was carried out under the principles established in the Helsinki Declaration. Additionally, this study received approval from the Ethics Committee of the Health Sciences Center at the University of Guadalajara (reference number #CI–07218).

## 3. Results

### 3.1. General Characteristics of the Study Population

The study population consisted of 260 patients with HCV infection (194 were HCV alone, and 66 were HIV/HCV coinfected). The proportion of men was higher in the HIV/HCV group than in the HCV group (86.4%, 57/66 vs. 38.7%, 75/194, *p* < 0.001). Additionally, the median age was higher in the HCV group than in the HIV/HCV patients (53.0 years, IQR: 44.0–59.0 vs. 40.5 years, IQR: 35.2–46.0, *p* < 0.001) (Table 1). Differences in risk factors were also found between the two groups. In the HIV/HCV group, intravenous drug use (IDU), tattoos, and piercings were more prevalent, whereas blood transfusions and surgeries were more common in the HCV group (Table 1).

### 3.2. Frequency of HCV Subtypes by Risk Factors

In this study, the most frequent HCV subtype was 1a (58.5%, 152/260), followed by subtypes 1b (19.2%, 50/260), 3a (13.1%, 34/260), 2b (5.4%, 14/260), 2a/2c (2.7%, 7/260), 2a (0.8%, 2/260), and 4a (0.4%, 1/260) (Figure 2A). Subtype 1b was more prominent in the HCV group than HIV/HCV (22.2%, 43/194 vs. 10.6%, 7/66, *p* = 0.040) (Table 1). HCV subtypes 1a and 1b were predominant among IDUs (70.4%, 38/54) and hemodialysis patients (60.0%, 3/5), respectively (Figure 2B). Additionally, HCV subtype 3a was prevalent among individuals with piercings (21%, 5/24), IDUs (17%, 9/54), and those with tattoos (16%, 14/89).

Among the less common HCV subtypes, 2b was found in health workers (17.0%, 1/6), individuals with piercings (12.5%, 3/24), and sex workers (11.1%, 1/9). HCV subtype 2a/2c was detected in 5.9% (2/34) of those receiving acupuncture, 4.0% (6/151) of individuals who had blood transfusions, and 3.7% (7/189) of surgical patients. HCV subtype 2a was present in 20.0% (1/5) of hemodialysis patients, 4.2% (1/24) of individuals with piercings, and 3.7% (1/27) of imprisoned individuals. HCV subtype 4a was identified in an individual with a history of imprisonment, intravenous drug use, and tattoos.

### 3.3. Risk Factors Associated with HCV Subtypes

This study also assessed the relationship among several risk factors and HCV subtypes (Figure 3). IDUs tended to have a 1.9-fold (95% CI = 1.01–3.71; *p* = 0.053) higher risk of HCV subtype 1a infection than non-IDUs. However, this factor did not increase the risk for HCV subtypes 1b or 2 (Figure 3B,C). Based on this observation, we performed a new analysis grouping HCV subtypes 1a or 3a versus 1b or 2. This analysis revealed that IDUs increased 3.22-fold (95% CI = 1.41–7.98, *p* = 0.004) the risk of HCV subtype 1a or 3a infection. Similar results were found with tattoos, which increased the 2.06-fold (95% CI = 1.12–3.94, *p* = 0.019) risk of HCV subtypes 1a or 3a. 

For HCV subtype 1b, the factors linked to its transmission were hemodialysis (Odds Ratio (OR) = 6.67; 95% CI = 1.08–51.73; *p* = 0.041) and blood transfusion (OR = 1.90; 95% CI = 0.97–3.94; *p* = 0.071) (Figure 3B). Identifying the factors that increase the risk of HCV subtype 2 was impossible due to the small sample size. Meanwhile, the factors that tended to increase the risk of HCV subtype 3a included IDUs (OR = 1.43, 95% CI = 0.60–3.21, *p* = 0.397), tattooing (OR = 1.4, 95% CI = 0.65–2.93, *p* = 0.381), and piercing (OR = 1.86, 95% CI = 0.58–5.07, *p* = 0.252).

### 3.4. HCV Subtype 1a Evolution in Mexico

Given the high prevalence and significant public health impact of HCV subtype 1a, we conducted an evolutionary analysis (Figure 4A). Based on the TMRCA, HCV subtype 1a may have been introduced to Mexico in 1976 (95% HPD = 1923–2004). Initially, the epidemic began with a few infections, with a median of 24 cases (Figure 4B). This event was followed by a diversification event in 1985 and 1986, leading to the formation of two distinct clusters (Figure 4A). From 1998 to 2006, infections increased exponentially, from 16 to 2161 cases (Figure 4B). After this rapid growth, the infection stabilized, reaching a median of 3161 cases by 2014. The shape of the HCV subtype 1a epidemic was similar to the HCV incidence curve reported by the Mexican government from 2001 to 2019 (Figure 4C).

## 4. Discussion

This study identified five HCV subtypes (1a, 1b, 3a, 2a/2c, and 4a) that exhibited differences in risk factors, particularly between HCV subtypes 1a and 1b. These differences may be attributable to the historical context of their respective risk factors. In Mexico during the 1970s and 1980s, the lack of regulation in the blood market, inadequate sterilization methods, and the use of reusable medical equipment (such as glass syringes) likely contributed to the spread of HCV [10]. During this period, women were particularly vulnerable to iatrogenic HCV infection due to frequent blood transfusions during or after childbirth, including cesarean sections [15,24,25,26,27]. These events could partly explain why most of the HCV-only group was composed of adult women aged 53 years. A study indicated that before 1998, most blood donors with detectable HCV viral loads in Mexico had HCV subtype 1b [28]. In this study, blood transfusions and hemodialysis were the main factors linked to HCV subtype 1b. Furthermore, HCV subtype 1b, blood transfusions, and surgeries were significantly more frequent in patients with HCV alone than in the HIV/HCV-coinfected group. These data suggest that the first outbreak of HCV in Mexico may have been caused by HCV subtype 1b related to medical procedures. However, due to the low number of subtype 1b samples, we could not precisely pinpoint the time of the most common ancestor of this subtype to reinforce the explanation above. Further molecular studies with subtype 1b samples are required to prove this hypothesis factually.

Furthermore, in this study, IDU was the main risk factor linked to HCV subtype 1a. This finding may be related to the implementation of sterilization stations in hospitals, the prohibition of blood sales, and the serological diagnosis of HCV in Mexico [10] that partially controlled the HCV epidemic due to subtype 1b until IDU emerged. IDU began in 1940 with the use of morphine (derived from the opium plant) for the treatment of pain [29]. During this period, IDU was not generalized nationwide; mainly, it was dominated by Sinaloa, Chihuahua, and Durango merchants, who cultivated opium in the western Sierra Madre and supplied it to America’s opiate market during World War II [29]. In 1947, one of the most sophisticated laboratories that transformed morphine into heroin was established in Guadalajara, Mexico [29]. At the end of World War II, the demand for morphine and heroin in the United States decreased, leading merchants to sell these substances within the Mexican market, causing an increase in drug consumption from 1965 to 1975 [30]. This situation was further complicated by the HIV outbreak in 1981 [31,32]. 

Additionally, IDU, tattoos, and piercings were more common in males with HIV/HCV coinfection than HCV alone. These results suggest that Mexico has experienced a gradual transition from HCV subtype 1b to 1a, attributed to the increase in the use of IDU and the decrease in the risk of HCV infection through blood transfusions, and the generation of patients infected with subtype 1b has declined. Spain has documented a similar change from 1b to 1a between 1988 and 2015 [33]. In France, it was reported that the new epidemic subtypes (1a and 3) associated with IDU replaced epidemic subtype 1a and endemic genotype 2 [34]. Likewise, genotype 3a emerged as the third most common in this research, and its association with percutaneous transmission of HCV is consistent with worldwide reports and earlier national studies [17]. 

The results of this study suggest that HCV subtype 1a could have been introduced to Mexico near 1976 (95% HPD = 1923–2004). This introduction likely resulted from complex migration patterns between Mexico and the United States. In 1942, the United States government proposed the Mexican Farm Labor Program, which promoted the arrival of 4.6 million Mexican men (Braceros) to work legally with temporary contracts in agriculture [35,36]. During wartime, the bracero program was crucial in supplying food to the United States and troops in war zones [37]. With the end of the Bracero program in 1964, many Mexicans lost their jobs, and some enlisted in the United States Army to participate in the Vietnam War, motivated by the possibility of obtaining American citizenship after service [35,38]. Approximately 80,000 Hispanics served in the Vietnam War [39]. During the war, about half of the soldiers were treated with opioids, such as heroin, to manage pain caused by combat-related injuries [40]. By the end of the war in 1975, between 7% and 15% of veterans had returned with signs of heroin dependence [40,41]. This post-war migratory pattern may have facilitated the spread of several pathogens to Mexico, including HCV subtype 1a. The diversification of HCV subtype 1a into two distinct clusters around 1985 and 1986 could be linked to the fast growth of the HIV epidemic in Mexico during the same period (1981–1990) [42,43]. This connection arises because both HCV and HIV share transmission routes, such as IDU and unprotected sex [44]. The diversification may have occurred in response to exposure to two different host environments: one characterized by HCV-monoinfected heterosexuals and the other by homosexuals infected with HIV. However, future studies are necessary to evaluate the evolutionary behavior of HCV subtype 1a within these two risk groups.

In the study, we found an exponential increase in HCV subtype 1a infections from 1998 to 2006, consistent with the incidence curve of HCV infection reported by the Mexican Health Secretariat [45]. Several factors could explain this increase. One significant factor is the shift in drug trafficking routes in the 1990s and early 2000s due to intensified Drug Enforcement Administration (DEA) control efforts in Colombia and the Caribbean [36]. Consequently, Mexico became a key transit point for drugs destined for the United States, increasing the availability of drugs within the country [46]. During this period, Mexican drug cartels significantly strengthened their influence and control over the production, distribution, and sale of drugs, facilitating greater access to drugs in the Mexican domestic market as well as in the United States and Europe [46,47,48,49]. This increased availability of drugs likely led to a rise in IDU, contributing to the observed rise in HCV subtype 1a infections.

This research provides relevant data at the regional level to improve the understanding of the risk factors and evolutionary history of subtype 1a, constituting an important record that captures the historical dynamics. Additionally, this study highlights the prevalence and characteristics of the hepatitis C Virus subtype 1a among drug users, which is crucial information. These findings are essential for the formulation of public policies aimed at mitigating the harm of drug trafficking, not only because of the security and public health issues it triggers but also because of the risk of hepatitis C infection among users. Nonetheless, several limitations must be considered. Firstly, the HCV subtypes found in this study may not reflect the full range of genetic diversity across Mexico due to its vast geographical and demographic heterogeneity. Secondly, this study focuses exclusively on the NS5B region of HCV, which is useful for phylogenetic studies but may not capture the entire genetic diversity of the virus. Additionally, the study’s sample size was relatively small, which could impact the robustness and generalizability of the findings. Like all cross-sectional studies, our research did not establish a cause-and-effect relationship between the identified factors and outcomes, emphasizing the need for cautious interpretation of the results. Further molecular epidemiology studies are required nationwide to gain insight into the routes of HCV transmission and the impact of prevention and elimination strategies.

## 5. Conclusions

This study found that HCV subtypes follow distinct transmission routes, emphasizing the need for targeted prevention strategies. Additionally, the findings provide valuable insights into the origin of HCV subtype 1a. By analyzing the history, risk factors, and dynamics of the HCV epidemic, we have identified these measures: limiting the harm of intravenous drug trafficking, enhancing medical training and infrastructure, and ensuring universal access to antiviral treatments. The successful implementation of these strategies could lead to an HCV-free future in Mexico.

## Figures and Tables

**Figure 1 viruses-16-01259-f001:**
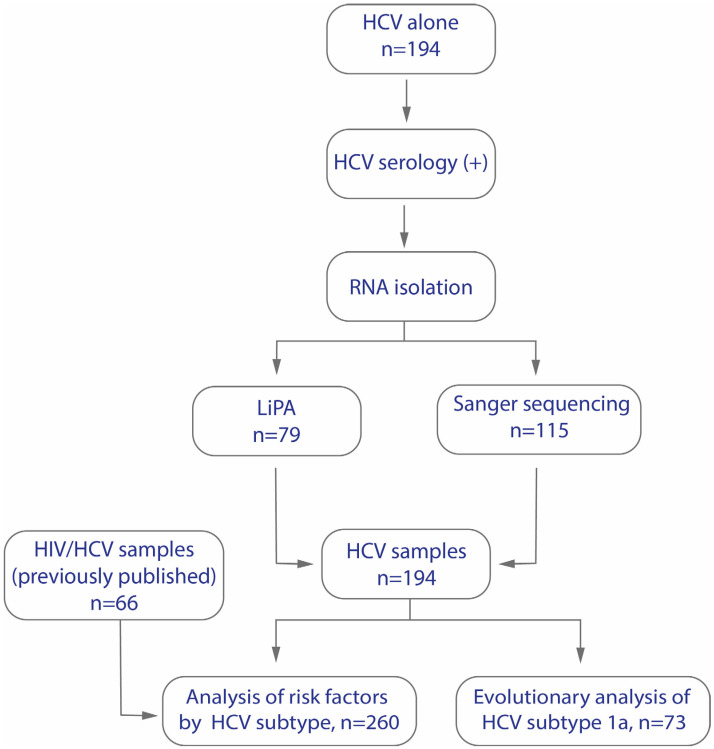
Sample collection strategy for studying risk factors, HCV subtypes, and the evolutionary analysis of HCV subtype 1a in Mexico.

**Figure 2 viruses-16-01259-f002:**
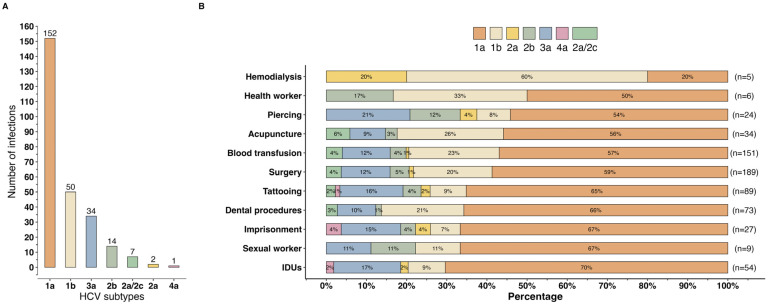
General distribution of HCV subtypes (**A**) and their frequency by risk factor (**B**). IDUs: Injection drug use.

**Figure 3 viruses-16-01259-f003:**
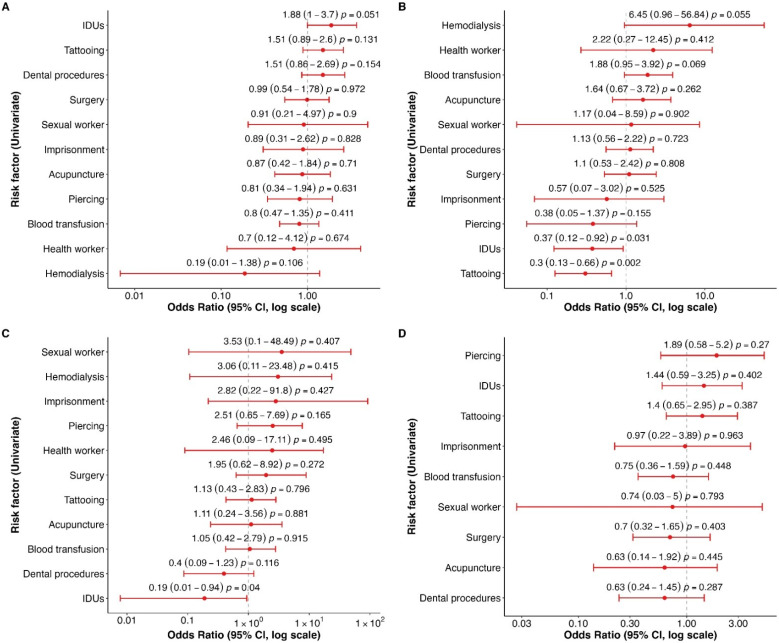
Univariate analysis was conducted to identify risk factors associated with HCV subtypes 1a (**A**), 1b (**B**), 2 (**C**), and 3a (**D**). IDUs: Injection drug users. The dots represent the Odds Ratio, and the bars indicate the 95% confidence interval.

**Figure 4 viruses-16-01259-f004:**
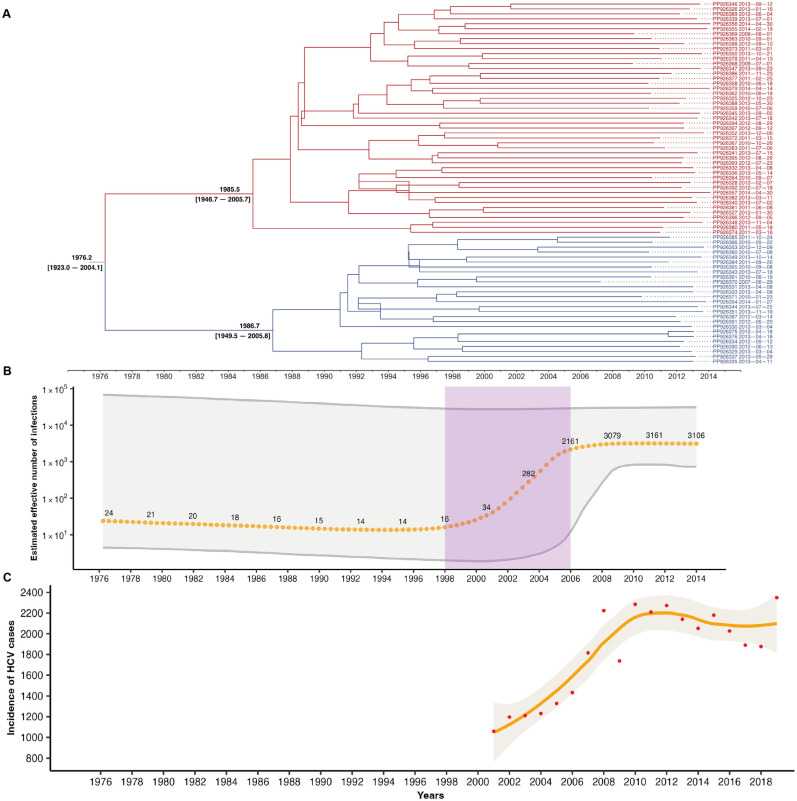
(**A**) The phylogenetic tree illustrates the evolutionary history of HCV subtype 1a infections in Mexico. The time to the most recent common ancestor (TMRCA) of HCV subtype 1a is at the tree’s base. The length of each branch represents the time elapsed from the root to the present. (**B**) Reconstruction of the HCV subtype 1a epidemic in Mexico from 1976 to 2014. The orange dots indicate the median estimated number of infections per year, and the gray shaded area represents the 95% highest posterior density interval. The purple rectangle highlights a phase of increasing HCV subtype 1a infections. (**C**) Graph showing the documented cases of HCV infections in Mexico from 2001 to 2019. The red dots indicate the number of infections per year, the orange line shows the trend, and the shade indicates the confidence interval of the trend line. These data were collected from Mexico’s Annual Morbidity Reports. Available at: http://epidemiologia.salud.gob.mx, consulted on 16 June 2024.

**Table 1 viruses-16-01259-t001:** General characteristics and risk factors of 260 HCV-subtyped Mexican patients.

Variable	Category	HIV/HCV (*N* = 66)	HCV (*N* = 194)	*p*-Value
Gender	Female	9 (13.6%)	119 (61.3%)	
	Male	57 (86.4%)	75 (38.7%)	<0.001 ^1^
Age (year)	Median (Q1, Q3)	40.5 (35.2, 46.0)	53.0 (44.0, 59.0)	<0.001 ^2^
Injection drug use	Absent	38 (57.6%)	158 (85.9%)	
	Present	28 (42.4%)	26 (14.1%)	<0.001 ^1^
Blood transfusion	Absent	37 (56.1%)	61 (33.3%)	
	Present	29 (43.9%)	122 (66.7%)	0.001 ^1^
Surgery	Absent	28 (42.4%)	33 (17.9%)	
	Present	38 (57.6%)	151 (82.1%)	<0.001 ^1^
Tattooing	Absent	27 (40.9%)	134 (72.8%)	
	Present	39 (59.1%)	50 (27.2%)	<0.001 ^1^
Piercing	Absent	46 (69.7%)	180 (97.8%)	
	Present	20 (30.3%)	4 (2.2%)	<0.001 ^1^
Dental procedures	Absent	43 (65.2%)	134 (72.8%)	
	Present	23 (34.8%)	50 (27.2%)	0.239 ^1^
Hemodialysis	Absent	65 (98.5%)	180 (97.8%)	
	Present	1 (1.5%)	4 (2.2%)	0.743 ^1^
Acupuncture	Absent	61 (92.4%)	155 (84.2%)	
	Present	5 (7.6%)	29 (15.8%)	0.096 ^1^
Imprisonment	Absent	39 (59.1%)	NA	
	Present	27 (40.9%)	NA	
Health worker	Absent	65 (98.5%)	179 (97.3%)	
	Present	1 (1.5%)	5 (2.7%)	0.584 ^1^
Sexual worker	Absent	57 (86.4%)	NA	
	Present	9 (13.6%)	NA	
HCV viral load (IU/mL)	Median (Q1, Q3)	4,480,000.0 (941,500.0, 15,625,000.0)	2,925,000.0 (680,000.0, 10,300,000.0)	0.331 ^2^
HCV subtyping method	LiPA	31 (47.0%)	79 (40.7%)	
	Sanger sequencing	35 (53.0%)	115 (59.3%)	0.375 ^1^
HCV subtype	1a	45 (68.2%)	107 (55.2%)	0.064 ^1^
	1b	7 (10.6%)	43 (22.2%)	0.040 ^1^
	2a	1 (1.5%)	1 (0.5%)	0.422 ^1^
	2b	2 (3.0%)	12 (6.2%)	0.327 ^1^
	3a	10 (15.2%)	24 (12.4%)	0.563 ^1^
	4a	1 (1.5%)	0 (0.0%)	0.086 ^1^
	2a/2c	0 (0.0%)	7 (3.6%)	0.118 ^1^

^1^ Chi-squared or Fisher’s exact test; ^2^ Mann–Whitney test; LiPA: Line Probe Assay; NA: Not available.

## Data Availability

The original contributions presented in the study are included in the article/Appendix A, further inquiries can be directed to the corresponding author.

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
