# Peer review of "Risk Factors Associated with Hepatitis C Subtypes and the Evolutionary History of Subtype 1a in Mexico"

_viruses, 2024, doi:10.3390/v16081259_

Round 1
Reviewer 1 Report
Comments and Suggestions for Authors
The specific study has significant limitations, which are described by the authors themselves in the last paragraph of the discussion. The main disadvantage concerns the small number of cases analyzed, thereby reducing the robustness of the results obtained. Additionally, this data size may not reflect the geographical heterogeneity of the state of Mexico. However, it constitutes an important record that captures developments in historical dynamics that should be communicated to the reader.
However, this study contributes little to the clinical practice and to the understanding of the epidemiological burden of this specific region of the planet. Similar phylogenetic changes have been observed in other regions and continents and have been documented in the literature by previous researchers. Nonetheless, every effort should be encouraged and applauded but in my opinion, it should not be published.
Best regards
Reviewer 2 Report
Comments and Suggestions for Authors
Title section, abstract, and introduction:
The title seems appropriate. Authors should consider providing their ORCiD numbers. Authors should consider providing an institutional email for correspondence.
I noticed that the abstract does not dwell on the specific conclusions (the text is too generalistic), and that may be a missed opportunity that the authors may want to reconsider.
Material and methods:
I will focus my comments on the phylogenetic analysis, which seems to be where I can make the best contributions. Moreover, phylogenetic methodology and results were a large factor that contributed to my scoring the quality of presentation, scientific soundness, and overall merit as average instead of high. The authors should modify the manuscript so that all of the following questions can be answered after reading it.
1) Can the alignments be made available to the reader? Were there InDels or ambiguities? How do the authors expect that InDels and missing data could impact this particular analysis, knowing that the selected strategy is sensitive to missing data and treats all gaps as ambiguities?
2) What were the parameters used for model selection? Have the authors considered using more robust and complete strategies for model selection, such as those implemented in IQ-Tree?
3) What is the justification for choosing posterior probabilities as an optimality criterion? How were priors selected? Can the authors provide detailed configuration files and scripts to facilitate the reproduction of this analysis? What criteria, exactly, did the authors use to decide that the analysis stabilized before being interrupted?
4) How were the molecular clocks calibrated? Why was that strategy chosen?
Figures:
Figure 1 needs minor editing. Everything written in the image should be readable. The unreadable text should be magnified or removed. The boxes on the bottom seem to be cropped. A chromatogram might be a better way to visually represent Sanger sequencing.
Figure 2 has very small text. Perhaps the authors would consider orienting the panel vertically, with Fig. 1a on top of Fig. 1b.
Terminals in Figure 3a must be enlarged to the point that they are readable.
Supplementary materials:
The authors must increase their efforts to share supplementary digital material. For example, for phylogenetic analysis, supplementary digital material may include scripts, alignment, and trees in text format (such as Newick or Nexus).
Supplementary digital materials could immensely contribute to this publication, elevating it to the point that it becomes easily reproducible.
Results and discussion:
There is an opportunity for the authors to expand the comparison of their results with the specialized literature. For example, this study encountered differences in risk factors by group. In the HIV/HCV group, intravenous drug use (IDU), tattoos, and piercings were more prevalent, whereas blood transfusions and surgeries were more common in 152 the HCV group
Conclusions:
My biggest concern with the manuscript in its present form is the proposal that "eradicating intravenous drug trafficking" is a key measure to be taken to significantly re-293 duce HCV infection rates. It is true that the results placed intravenous drug use (IDU) as the main risk factor linked to HCV subtype 1a. But the authors took a leap from there to the response: eradication. This study can say that IDU is a major risk factor, but it does not have the data to support eradication as the best solution to tackle this problem. I recommend that the authors should be more careful in crafting their conclusions to guarantee that they are based only on their results.
Comments on the Quality of English LanguageThe text is well written, and I won't list the minor changes required. I recommend the authors give it another pass during this review stage to find and fix minor typos. Also, it is easy to identify long sentences that are not in the direct order. Making those shorter and more direct can make the text slightly repetitive but would make it less ambiguous in some points and easier to understand. Overall, however, the language is good.
Reviewer 3 Report
Comments and Suggestions for Authors
While the association of the risk factors with HCV subtypes was nicely done, the robustness and description of the Bayesian statistical methods used for the evolutionary history reconstruction need significant improvements. I highlight the major concerns and recommendations as follows:
1. The bigger the sample size, the better the data output. Have you considered increasing the GT1a samples with the HCV/HIV samples (assuming some are GT1a)?
2a. HCV genotyping/subtyping must be confirmed by phylogenetic analysis with the addition of relevant reference controls (the LANL HIV/HCV database has curated reference controls). A maximum likelihood method will be good enough for this purpose, and the rooted ML tree as proof of genotype/subtyping classification (can be a stand-alone figure or supplementary figure). This aspect was alluded to in the methods section although with NJ method which is a very simplistic tree-reconstruction method (ln 112 -113) but there is no evidence that this was done in this study.
2b. HCV GT1a forms two distinct clades, I and II and that might be evident in Fig 4A (colour coding of the two clades is missing). Did you further classify these samples according to these clades (Geno2pheno)?
3. Transmission cluster analysis is something else to consider with the ML tree. You could evaluate how the clusters associate with transmission routes and annotate that in the ML tree.
4. Bayesian coalescent analysis as described in the methods section, lacks clarity on the priors and sufficient quality checks for reproducibility purposes.
First, how many independent MCMC runs were run for this analysis? And how did you confirm the convergence of chains between runs?
Second, I am surprised that TN93 was the best substitution model. Was this based on AIC or Bayesian information criterion (BIC)?
Third, did you test for the molecular clock likelihood of your dataset? Did you estimate the evolutionary rate, and how does it compare to other studies?
Fourth, what does this mean "The best-fitting model was selected based on the Akaike information Criteria (AIC) with 1000 replicates" (ln 123-124)? Did you perform Marginal likelihood estimation of both clock models? What is the resulting Bayes factor resolution?
5. Based on the historical context regarding the earlier incidence of GT1b (ln 218 - 219), you surmised that the first outbreak may have been caused by GT1b (ln 222 - 224), is there any phylogenetic evidence to support this claim? How the paragraph (ln 238 - 246) was framed suggests an evolutionary history of the HCV subtypes in Mexico placing GT1b older than GT1a and GT3, which I find misleading. Why don't you phylogenetically resolve the evolutionary history of GT1b and GT3 to support this conjecture?
I recommend this paper to guide you in addressing most of these concerns: https://doi.org/10.1038/s41598-020-69692-7
Round 2
Reviewer 1 Report
Comments and Suggestions for Authors
No additional comments
Author Response
Dear Revisor !,
Thank you for your comments.
Reviewer 3 Report
Comments and Suggestions for Authors
You did not address the question regarding the molecular clock likelihood of your dataset. Can you confirm your dataset has a sufficient temporal signal for BEAST analysis?
AIC/bootstrapping (ML/frequentist statistics) of posterior probability (Bayesian statistics) is not ideal, and one should be cautious with its interpretation. Bayes factor is a more appropriate approach under the Bayesian framework.
Author Response
Dear Revisor 3,
Please find the replies to your two comments.
A1. Thank you for your comment and the opportunity to clarify this point. We can confirm that our dataset has sufficient temporal signal for BEAST analysis. We used the TempEst program to perform a root-to-tip regression analysis, and the results indicate that the temporal signal is adequate for the molecular clock approach in BEAST. Page 4, lines 124-125.
A2. We understand and share your concern regarding the need to be cautious with the interpretation of results obtained through AIC/bootstrapping (ML/frequentist statistics)
We want to point out that our results are not solely based on these statistical methods but are also supported by historical epidemiological data detailed in the analysis. This data provides additional context and validation, which strengthens our conclusions. We recognize that using the Bayes factor is a more appropriate approach within the Bayesian framework, and we are considering its implementation in future analyses.
